Summer pruning in apple trees is an advisable cultural practice that promotes bud differentiation and improves fruit quality: a literature review

Qiu Xiaoyu 1
Xing Wenxi 2
Que Youxiong queyouxiong@126.com 3
Lu Guilong luguilong666@126.com 4
1 School of Plant Protection and Environment, Henan Institute of Science and Technology , Xinxiang , Henan , China
2 Institute of Forestry, Sichuan Academy of Forestry Sciences , Chengdu , Sichuan , China
3 State Key Laboratory of Tropical Crop Breeding, Institute of Tropical Bioscience and Biotechnology, Sanya Research Institute, Chinese Academy of Tropical Agricultural Sciences , Sanya , Hainan , China
4 School of Horticulture and Landscape Architecture, Henan Institute of Science and Technology , Xinxiang , Henan , China
Brygadyrenko Viktor
Electronic publication date: 2025 Oct 23
Publication date: 2025
Volume: 13
Electronic Location ID: e20229
Received 2025 Apr 22; Accepted 2025 Sep 23
Copyright: ©2025 Qiu et al.
Copyright year: 2025
Copyright holder: Qiu et al.
License: This is an open access article distributed under the terms of the Creative Commons Attribution License, which permits unrestricted use, distribution, reproduction and adaptation in any medium and for any purpose provided that it is properly attributed. For attribution, the original author(s), title, publication source (PeerJ) and either DOI or URL of the article must be cited.
License URL: https://creativecommons.org/licenses/by/4.0/

Keywords: Apple, Summer pruning, Nutritive and reproductive growth, Bud differentiation, Fruit quality

Funding: Scientific Research Start-up Fund for High-level Introduced Talents of Henan Institute of Science and Technology 103020224001/073 This work was supported by the Scientific Research Start-up Fund for High-level Introduced Talents of Henan Institute of Science and Technology (103020224001/073). The funders had no role in study design, data collection and analysis, decision to publish, or preparation of the manuscript.

==============================
Apples (Malus domestica) are among the most widely cultivated and economically valuable fruit tree crops worldwide. Summer pruning, encompassing thinning, branch bending, and ring wounding, is a method of regulating apple production and is important to fruit tree growth. Timely and appropriate application of this measure can control the vigorous growth of new shoots, promoting the differentiation of flower buds, enhancing early fruiting and yield of young trees, improving the ventilation and light conditions of trees, and reducing the occurrence of pests and diseases, thereby achieving superior fruit quality. However, excessive pruning negatively affects the tree’s strength, yield, and fruit quality. In this review, we concisely describe the physiological basis of summer pruning, the main manual pruning methods, the latest robotic pruning technologies, and their specific impacts on fruit quality and yield. We also analyze the main problems in current production, emphasizing the importance of robotic pruning as an important management measure for future large-scale and intelligent development of the apple industry. This information is beneficial to fruit tree researchers and growers and provides a scientific reference for apple production.

INTRODUCTION

Apples (Malus domestica) originated from the Xinjiang region of China and Central Asia (Duan et al., 2017; Brite, 2021). Their domestication and cultivation history can be traced back to 4,000–10,000 years ago (Cornille et al., 2014). Apple fruits are rich in nutrients and can be eaten fresh or converted into juice, wine, or preserved fruit. Moreover, apples rich in phenolic substances can be used to process health products after extraction (Rana et al., 2021). Apple trees, such as B9, M9T337, M26, and those used as rootstocks, have various uses (Wang et al., 2019). The species Malus toringoides (Rehd) Hughes and Malus transitoria (Batalin) C.K. Schneid are ornamental trees, and young leaves or tender shoots can be processed into Tibetan tea in China (Lu et al., 2022). Owing to their strong adaptability, apple trees are now commonly grown worldwide, from temperate regions to the cold climate of Siberia (Chen et al., 2021a), and are distributed in high-altitude areas of tropical and subtropical regions (Ashebir et al., 2010). The total area for apple cultivation is approximately five million hectares, with apple production exceeding 95 million tons, generating an annual gross product of nearly USD 50 billion. Among apple-producing countries, China ranks first in terms of cultivation area and production quantity, accounting for 44% and 50%, respectively. However, China exports 0.8 million tons, representing only 13% of the world’s total exports. Furthermore, the current average yield of apples in China is 22.3 t/ha, which is slightly higher than the world average (19.9 t/ha) but much lower than that of developed agricultural countries (approximately 33–51 t/ha), such as Chile, France, Italy, Israel, and the United States (Food and Agriculture Organization of the United Nations (FAO), 2023).

The most vigorous growth period for apple trees is summer: June–August in the Northern Hemisphere and December–February in the Southern Hemisphere (Kuden et al., 2023; Lugaresi et al., 2024). During this period, rapid growth and high branching can easily cause depressions in trees, between fruiting branches, and in the interior, which may be accompanied by growth disorders, ineffective nutrient consumption, and pest and disease outbreaks that can reduce flower buds and fruit quality (Liang et al., 2022). However, orchard management focuses mainly on winter pruning and ignores summer pruning, hindering fruit trees from reaching their full potential in terms of both quality and quantity. In addition, farmers’ experience with summer pruning is insufficient, particularly among small-scale farmers in China (Jin et al., 2022). In recent decades, apple cultivation systems have evolved from arbor and sparse planting to dwarf and close planting, which has become a trend in the apple industry. Timely summer pruning is critical for modern apple production because the growth of densely planted orchards is more compact than that with traditional methods (Gandev & Dzhuvınov, 2014).

Current research on summer pruning is relatively limited compared with that on dormant pruning. Reasonable application of these measures can improve canopy light (Chaploutskyi, 2024), reduce the occurrence of pests and diseases (Cooley, Gamble & Autio, 1997; Maughan, Black & Roper, 2017; Wallis, Miranda-Sazo & Cox, 2021; Zhang et al., 2024a), and help to obtain the best fruit quality (Reig et al., 2019); it can also promote young tree shaping and control new shoot growth, with the effects being most significant in tall spindle- and pillar-shaped apple orchards (Ashraf & Ashraf, 2014; Anthony, Serra & Musacchi, 2020; Rehman, Singh & Saxena, 2021; Nasrabadi et al., 2022). Determining the effectiveness of summer pruning would provide a scientific and practical basis for enhancing flower bud differentiation (Xing et al., 2016; Vosnjak, Mrzlic & Usenik, 2022; Zhang et al., 2023). For a clearer understanding of summer pruning of apple trees, this review provides an overview of the physiological basis of summer pruning and pruning methods as well as their effects on fruit quality and yield. It also highlights current challenges of this practice and explores its future in the apple industry.

SURVEY METHODOLOGY

To ensure comprehensive and unbiased coverage of the literature, this study focused on publications from the year 2000 onward, except for a few basic articles, with no geographical location limitation. The document types included articles, reviews, and conference proceedings, retrieved primarily from PubMed, Google Scholar, Springer, Web of Science, China National Knowledge Infrastructure, and the Wanfang Database. To be precise in our search, several terms and phrases were used, such as “summer pruning”, “manual pruning”, “robot pruning”, “machine pruning”, “flower bud differentiation”, “fruit quality”, and “cultivation management” in apple trees or related industry. Studies related to genetics, environmental factor, and non-pruning interventions were excluded (Fig. 1). In particular, our years of experience in the summer pruning of apple trees have been integrated into the present review. In addition, the area and yield data of apple from Food and Agriculture Organization of the United Nations was also used as a supplementary analysis (Fig. 2).

Figure 1 PRISMA flow for summer pruning in apple trees.

The statistical data comes from searching various databases, PubMed (https://pubmed.ncbi.nlm.nih.gov/), Google Scholar (https://scholar.google.com/), Springer (https://link.springer.com/), Web of Science (https://www.webofscience.com/), China National Knowledge Infrastructure (https://www.cnki.net/), Wanfang Data (http://www.wanfangdata.com/).

Figure 2 Apple cultivation area and production in the world and China.

The data is sourced from FAO Statistical Databases (FAOSTAT). Available at http://www.fao.org/faostat/zh/#data.

PHYSIOLOGICAL BASIS OF SUMMER PRUNING IN APPLE TREES

The physiological differentiation of apple flower buds mainly occurs from early June to July in the Northern Hemisphere. When nutrients begin to accumulate after new shoot growth stops, and the production of hormonal substances is conducive to flower formation, there are a series of physiological and biochemical transformations from vegetative to reproductive growth within the growing point (Zhang, 2011; Ashraf et al., 2020). For more than a century, botanists have proposed several hypotheses, such as the carbon-nitrogen ratio theory, endogenous hormone balance, critical node number, and gene initiation, to elucidate the potential mechanism of physiological differentiation (Jin et al., 2023). The basic processes of flower bud differentiation in fruit trees are as follows: The growth point is composed of a homogeneous population of cells in the proto-differentiated tissues, all of which are genetically totipotent, and not all genes show activity during any period in the cells. Only when certain substances (flower-inducing hormones) are produced under the combined effect of external conditions (e.g., light, temperature, and water) and internal factors (e.g., changes in hormone ratios and accumulation of energetic substances) can flower-inducing genes be activated, resulting in changes in enzyme activities and hormones and high-intensity uptake of nutrients, ultimately leading to morphological differentiation in flower buds (Zhang, 2011; Xing et al., 2015; Chen et al., 2018; Izawa, 2021; Milyaev et al., 2021).

Before the morphological transformation of vegetative buds into flower buds in fruit trees, the physiological and biochemical states during the growth stage, as well as the direction of metabolic pathways, are unstable. Therefore, physiological differentiation, which is essential for regulating bud differentiation, is critical for floral induction. Once bud morphological differentiation begins, it continues stepwise. Morphological differentiation of flower buds begins 1–7 weeks after physiological differentiation (Wilkie, Sedgley & Olesen, 2008). Coen & Meyerowitz (1991) and Weigel & Meyerowitz (1994) proposed and refined the “ABC” model for determining the characteristics of floral organs; namely, there are three types of genes with different functions during flower bud development whose joint action determines the direction of the development of floral meristems. Previous studies have confirmed that APETALA2 (Wollmann et al., 2010), FLC (Xu, Tao & He, 2022), FT/TFL1 (Koutina & Pepelyankov, 2010; Wickland & Hanzawa, 2015), MdAGL24-like (Su et al., 2018), MdGRF11 (Zuo et al., 2022), MdIPT1 (Jia et al., 2023a), and MdBLH14 (Jia et al., 2023b) are involved in flower bud differentiation. Additionally, multiple miRNAs, including miR156, miR159, miR160, miR164, miR166/165, miR167, miR169, miR172, miR319, miR390, and miR399, are involved in flower induction and formation (Hong & Jackson, 2015), whereas MdGAMYB (Zhang et al., 2024b) and MdWRKY71 (Su et al., 2025) are closely associated with the regulation of flowering time. A recent study demonstrated the fine-tuned regulation of flowering by the MdbHLH48-MdFT1/MdTFL1-MdWRKY6 module and provided insights into flower bud formation in apples (Zuo et al., 2024).

The quality of flower buds formed by fruit trees is closely related to their nutritional level, growth environment, and other factors (Koutinas, Pepelyankov & Lichev, 2010). Balancing the reproductive and vegetative growth of branches is the main method of controlling flower bud differentiation. Bending branches, twisting shoots, and other summer pruning measures alter the direction of branch growth and slow it. Consequently, starch accumulation in the branches increases, total nitrogen content and auxin levels decrease, and cytokinin levels increase (Xing et al., 2016; Xing et al., 2019). In addition, polyamine content is altered in branch buds to promote flower bud differentiation (Lv, Ma & Zhang, 2018) (Fig. 3). The proportion of flower buds increased as the bending angle increased from 70° to 110° (Zhang et al., 2017), with early bending being the most effective (Chen et al., 2023). Nutrients are the basis for flower bud differentiation, and pruning measures (e.g., ring wounding) that damage the cortex and xylem of branches and trees can prompt a concentrated flow of nutrients to the injured site, increasing cytokinin and decreasing gibberellin concentrations in injured branches (Zhang et al., 2021; Zhang et al., 2023). Elevated protein levels and reduced soluble sugar content in buds (Wang et al., 2021) are conducive for flower bud formation. Light is essential for flower bud formation, and appropriate thinning can improve light conditions, leaf photosynthetic efficiency, aboveground nutrient accumulation in trees, and bud quality (Mierowska et al., 2002; Vosnjak, Mrzlic & Usenik, 2022). The suitable temperature for apple bud differentiation is 20–27 °C. Altered climatic conditions, exemplified by increased spring warming coupled with sub-freezing temperatures, negatively affect flower bud development (Kumar et al., 2024) and may also affect flowering time (Maple et al., 2024).

Figure 3 Flower buds under different summer pruning treatments.

(A) Bending branches, (B) ring cutting to main branches, (C) ring cutting to the trunk, (D) rotating branches downward, and (E) flowering in the year after the twisting treatment. The picture was taken in the apple orchard of Yueduo Fruit Industry Co., Ltd. in Rongcheng City, Shandong Province.

MAIN METHODS OF SUMMER PRUNING IN APPLES

Manual pruning

Apple-tree production is typically performed manually in underdeveloped agricultural countries. Pruning accounts for approximately 20% of the total labor cost and is one of the most labor-intensive operations (Crassweller et al., 2020). Manual pruning relies primarily on the experience and skill of fruit farmers. Summer pruning, a management measure for apple production, is commonly used in orchards in China and other countries. Over many years of practical application, its methods and techniques have been gradually improved (Zhang, 2011; Zhang et al., 2017; Lv, Ma & Zhang, 2018; Xing et al., 2019; Wang et al., 2021; Zhang et al., 2021; Chen et al., 2023). The details are as follows.

Thinning branches. Lateral branches below the heading height, excessive water sprouts, and branches that are competing, diseased, or densely overlapping should be promptly removed (Maughan, Black & Roper, 2017; Robinson, 2025). The upper and lower branches of densely planted fruit trees should be maintained at approximately 30–40 cm in length. Notably, when the diameter of a main branch exceeds one-third the thickness of the main stem or is greater than two cm, the branch becomes thinner over time, and approximately one cm of the horse’s ear-slanting stakes should be retained to allow the buds underneath the stakes to sprout into new branches. Furthermore, large branches should be thinned sparingly, and trees should be removed and renewed by at most 2–3 branches per year to ensure fruit production and tree strength (Lu et al., 2020).

Bending branches. This pruning type is also known as “opening of the base angle”. Once the length of the main branch of a fruit tree exceeds 30 cm, the branch should be bent as early as possible to maintain an appropriate base angle of 90–110° (Han et al., 2008; Xing et al., 2016; Xing et al., 2019). Because vigorous growth and hormonal effects tend to promote upright branching in spring and this treatment is most effective in summer, the method can be subdivided into supporting, bracing, falling, pulling, and other methods based on the use of different tools (Fig. 4). Before bending, the branch is protected below the base by using one hand. The other hand gently kneads up and down to soften the branch from the base to the front with care and force to avoid breaking the branch or causing serious injury to the cortex. To avoid wounding the branches by strangulation, the tools (e.g., nylon rope, cloth strip, and iron wire) used to bend the branches should not be thin (Qiu & Sang, 2025).

Figure 4 Means of summer pruning in apple trees.

(A) Toothpick support, (B) angle opener support, (C) pulling down with a nylon rope, (D) twisting, (E) thinning, (F) ring cutting the branch, (G) ring cutting the main stem, and (H) ring stripping the main stem. The picture was taken in the apple orchard of Yueduo Fruit Industry Co., Ltd. in Rongcheng City, Shandong Province.

Twisting shoots. When fewer lateral branches are present in the main branch, the new upright branches can be twisted. When the length of a branch reaches 20 cm or more and is semi-woody, it can be twisted 180° from the base by 3–5 cm. The xylem and phloem are wounded (unbroken) and evenly distributed on both sides. In addition, twisting should be used in conjunction with bending and thinning. Its application in ‘Fuji’ apples is highly effective but is ineffective in ‘Red Delicious’ and ‘Jonagold’ apples; the method easily causes “dead-top” because new shoots of these varieties are slightly harder and brittle. As all apple tree varieties harden after spraying with paclobutrazol, they must be twisted before spraying (Zhang, 2011).

Rotating branches downward. Turning the branches downward can strengthen the inhibition of vegetative growth in large, fully lignified, strong, upright, and competitive branches. Before the treatment, the proposed growth direction of the branches is determined. The branches are gently pushed horizontally to the left or right several times, slowly twisted downward, and pressed to maintain a slightly downward head. The branch xylem is slowly twisted and slightly fractured to avoid breaking the branches with excessive force. In addition, the basal angle should be increased for strong branches to strengthen their inhibitory effects (Zhang et al., 2015).

Ring wounds and bud-notching. Ring cutting or stripping, cutting through, and partially or completely removing the narrow-width ring from the bark can be conducted on apple varieties with high budding power and strong trees, such as ‘Orin.’ Ring cutting or stripping should be performed cleanly and neatly, and should not be so deep that it cuts through the cortex to avoid injuring the xylem. The width of ring wounding is 1/10 to 1/5 of the diameter of the branch and should not exceed one cm, and the wounds should heal in approximately 20 days. For ‘Red Delicious’ lines, such as ‘Starking Red’ and ‘Starkrimson,’ in the case of weak growth, ring cutting instead of ring stripping should be applied (Zhang, 2011). Bud-notching, also known as scoring, involves cutting the cambium layer without removing the bark. The wound is made horizontally with a hacksaw blade 1–2 mm below the buds, and the flow of solutes through the phloem is stopped or slowed for a shorter period than that by ring stripping, avoiding massive dieback of the root system due to ring stripping (Maimaiti et al., 2013a; Close & Bound, 2017; Fallahi et al., 2019). Bud-notching of branches with a few buds can promote the formation of short fruit-bearing branches, particularly in varieties with low sprouting rates or poor branching power.

In addition to the aforementioned methods, the following summer pruning methods are commonly used: (1) Cutting back. When the branches are too long, such as tall spindle-shaped branches longer than 0.8 m, they shorten and retract, weakening their growth. (2) Shoot removal. Excess sprouts below the heading height and pruning cuts should be removed as early as possible to improve growth conditions in the tree canopy. (3) Top pinching. A fruit branch with a large hollow cavity can be stimulated to sprout lateral buds by top pinching. This method is usually performed on upright branches when they reach 15–20 cm in length and on extended branches when they reach 50–60 cm in length (Qiu & Sang, 2025).

Robotic pruning

While manual pruning is effective, it is labor-intensive and not conducive to large-scale development of the apple industry. To the best of our knowledge, the development and application of agricultural robots are changing the agricultural models (Shamshiri et al., 2025). Robotic pruning, often used in developed agricultural countries, is a potential long-term solution to address the increasing efficiency, labor shortages, and associated high costs (Mika, Buler & Treder, 2016; Zhang, Karkee & Tabb, 2019; Zahid et al., 2021a).

Robotic pruning involves the use of laser scanning, virtual reconstruction, virtual interaction, image acquisition, and other techniques to build a data model and apply it in combination with deep-learning techniques for fruit tree pruning work (Bai et al., 2019; Tong et al., 2022; You et al., 2022). Specifically, the pruning machinery has a high degree of adaptability and can dynamically adjust the position and angle of the pruning tool according to the information in the model. It can realize precise cutting of branches through an accurate control system to minimize damage to trees and improve the efficiency of pruning. Furthermore, a mechanical pruning system can integrate machine vision and artificial intelligence algorithms, enabling the machine to recognize the growth status of branches and fruit development in real time during the pruning process to make reasonable decisions (He & Schupp, 2018; Zahid et al., 2020; Zahid, Mahmud & He, 2021b; Zahid et al., 2022). For example, Karkee et al. (2014) pruned branches by constructing 3D skeletons of tall spindle apple trees, which removed 85% of the long branches and only 69% of the overlapping branches. Kang et al. (2025) employed an improved version of the YOLOv8-seg model to segment the trunk and primary branch regions from RGB images; the results showed that the accuracy of pruning decision judgment was 88.3%, the success rate of performing end pose estimation was 89.9%, and the total time of processing was 5.4 s per image. Zahid, Mahmud & He (2021b) evaluated the torque required for robotic apple tree pruning, and statistical analysis suggested that the torque required for pruning ‘Honeycrisp’ apple trees was significantly lower than that required for ‘Fuji’, ‘Gala’, and ‘Golden Delicious’ apple trees. Intelligent pruning can improve the accuracy of apple tree pruning and provide new ideas for standardizing and scaling the management in the apple industry (Zhang, Karkee & Khot, 2017; Wang, 2024). It can also be used as a reference for flower and fruit thinning (Dias, Tabb & Medeiros, 2018).

IMPACT OF SUMMER PRUNING ON FRUIT QUALITY AND YIELD OF APPLES

Proper summer pruning is essential for improving the postharvest quality of apples. Increased light transmission within the canopy following summer pruning promotes anthocyanin synthesis in apples, which results in uniformly and deeply colored fruit skin, significantly improving the appearance of apples, especially red-skinned and red-fleshed varieties (Autio & Greene, 1990; Musacchi & Serra, 2018; Kviklys et al., 2020; Chen et al., 2021b; Andergassen et al., 2023). Intrinsic quality indicators such as total phenolic content, antioxidant activity (Lugaresi et al., 2024), soluble solids (Bhusal, Han & Yoon, 2017), and flesh firmness (Lugaresi et al., 2022) are significantly increased in fruits from pruned trees, whereas acidity levels decrease (Ashraf et al., 2017). Notably, moderate summer pruning, which reduces the incidence of fruit pests and diseases (Sharma et al., 2018; Qiu & Sang, 2025), improves the ventilation and light transmission conditions in tree canopies, especially in some varieties with strong growth potential (e.g., Orin and Ralls Janet). Summer pruning lowers the risk of scab and powdery mildew (Holb & Kunz, 2016), with weak pruning associated with higher levels of apple scab infection, particularly in organic systems (Holb, 2005). In addition, pruning increases the Ca/K ratio in fruits and substantially reduces the incidence of bitter pits (Guerra et al., 2021) and flesh browning (Lugaresi et al., 2024). It can also improve storage tolerance and overall fruit quality (Guerra & Casquero, 2010). Furthermore, the timing of pruning considerably affects fruit quality; for example, Djordjević, Dejan & Gordan (2020) observed that earlier pruning was more favorable for fruit quality than later pruning.

Interestingly, moderate summer pruning enhances production efficiency (Robinson, Lakso & Ren, 1991; Vaibhav, 2023), light interception (Palmer, Avery & Wertheim, 1992), leaf chlorophyll content, blade thickness, photosynthetic efficiency, leaf weight (Maimaiti et al., 2013b), and nutrient transport to the fruit, thereby increasing single-fruit weight (Bhusal, Han & Yoon, 2017; Rehman et al., 2024). It also promotes flower bud formation, improves flower quality (Marini, Sherif & Smith, 2020), ensures yield in the following year, and to a certain extent, avoids the phenomenon of major or minor years (Platon & Zagrai, 1997; Campbell & Kalcsits, 2024). This practice also optimizes the environment for internal tree growth and increases water use efficiency (Ye et al., 2021) and soil respiration rates in the short term (3–10 days) (Glenn & Campostrini, 2011). During the growing season after summer pruning, lowering water consumption and improving the water status may benefit fruit growth and help offset potential detriment caused by carbohydrate deficits (Li et al., 2003a). Compared with unpruned trees, glucose and fructose levels generally increase in apple root sorbitol as the amount of summer pruning increases, with no effect on sucrose or phloridzin (Ferree et al., 1984). Uselis et al. (2021) observed a tendency of high yields with mechanical pruning. In addition, summer pruning increases the branch cross-sectional area (Seleshi, Mosie & Setu, 2023), thereby enhancing tree strength and resistance and extending the fruitfulness and fruiting period of the trees.

MAIN PROBLEMS IN CURRENT SUMMER PRUNING

In apple production, both summer and winter pruning are indispensable and complement each other (Campbell, Schupp & Marini, 2023). Generally, summer pruning increases labor costs to a certain extent and reduces the workload of winter pruning. The key to effective summer pruning is to perform it in a timely manner, in small amounts, and over multiple sessions. This helps the tree achieve a good structure with a fruiting branch hierarchy, reasonable spatial distribution, and sunlight exposure for all branches (Sharma et al., 2018; Blagoeva & Krishkova, 2020; Kumawat, Raja & Nabi, 2022). Pruning should minimize leaf area loss and branch damage to avoid damaging the root system, as the aboveground parts of a tree are closely related to the belowground parts (Li et al., 2003b). The timing, method, and frequency of summer pruning should be specific to fruit tree growth (Dahapute et al., 2018; Djordjević, Dejan & Gordan, 2020; Gizaw et al., 2023), and pruning must be avoided during rainy weather to prevent wound infections and tree weakening (Autio & Cooley, 2011; Chen et al., 2016).

Importantly, summer pruning requires a high level of expertise from orchard managers. When intense, summer pruning can reduce leaf area, leading to sunburned fruits (Gonda et al., 2006; Chen et al., 2021b), reduced fruit size, lower soluble solids and watercore content (Myers & Ferree, 1983), and delayed fruit maturity (Saure, 1987), all of which may be detrimental to the following year’s yield (Nasrabadi et al., 2022). The main problems that often exist in summer apple pruning are discussed below.

In manual pruning, most fruit farmers in underdeveloped agricultural countries have not undergone systematic training, and the consideration of varietal characteristics, tree shape, and tree strength is insufficient (Jin et al., 2022; Chaploutskyi et al., 2023), resulting in uneven pruning quality. Examples include the following: untimely branch bending, bending at a small angle, and bending in an unreasonable position; twisting early or late, resulting in severe branch damage; thinning or cutting back heavily, resulting in yield reduction in the second year; and leaving long stubbles at the time of thinning, resulting in excessive sprouting of new shoots (Zhang, 2011).

Not surprisingly, robotic pruning can significantly improve work efficiency and liberate labor; however, it still has limitations. The current developments in robotic pruning are more for winter pruning than for summer pruning, and the methods are relatively single, mainly for thinning and cutting back, ring wounds, and bending branches; other methods cannot be effectively completed. Moreover, accurate pruning in summer is affected by shading from leaves and fruits and internal light conditions. Tree training systems, pruning methods and strategies, 3D structural reconstruction of tree branches, and practical mechanisms or robotics require further research to develop effective tree-branch pruning systems (He & Schupp, 2018; Zahid et al. 2021a; Zahid, Mahmud & He, 2021b). Furthermore, the application of robotic pruning is limited by many factors, such as field terrain, worker skill level, planting scale, and input capacity. In China, apple orchards are mostly planted on hills and mountains and are mainly owned by smallholders (Mhamed, Kabir & Zhang, 2023), which limit the widespread application of robotic pruning. There is an urgent need to develop pruning robots that are simple to operate, multifunctional, and are miniaturized and highly adaptable (Fig. 5). The mechanisms underlying the effects of time and summer pruning methods on bud differentiation and fruit quality require further research to provide a scientific basis and guidance for production practices.

Figure 5 The main research content of robotic pruning.

CONCLUSIONS AND FUTURE PROSPECTS

Summer pruning is key to canopy shaping in the management of fruit trees and is an important initiative to promote early fruiting and higher yields in young apple trees. This environmentally friendly technique, which prevents and reduces the occurrence of pests and diseases, could be an ecological alternative to postharvest calcium treatment. With rising labor costs, the use of intelligent robotic pruning will increase in the future. Consequently, the integration of intensive planting with light pruning, intelligent, and precision pruning is emerging as a key future development trend in the apple industry. Moreover, from a long-term perspective, pressures and challenges are increasing. According to the United Nations, the global population is projected to exceed 9.7 billion by 2050 (UNDESA, 2022), indicating that the market demand for apples will increase further. Additionally, frequent global climate extremes, increasing chemical pollution, and decreasing available land resources have contributed to the instability of apple production and indirectly affected industrial security. Therefore, in future apple production, summer pruning coupled with water and fertilizer management should be conducted in a timely manner according to the growth of fruit trees. This approach can promote bud formation, improve fruit quality, and increase the yield in the following year. In conclusion, summer pruning is an advisable cultural practice in apple production.

Additional Information and Declarations

Competing Interests

Author Contributions

Data Availability

The authors declare there are no competing interests.

Xiaoyu Qiu conceived and designed the experiments, performed the experiments, analyzed the data, prepared figures and/or tables, authored or reviewed drafts of the article, and approved the final draft.

Wenxi Xing conceived and designed the experiments, performed the experiments, authored or reviewed drafts of the article, and approved the final draft.

Youxiong Que conceived and designed the experiments, performed the experiments, authored or reviewed drafts of the article, and approved the final draft.

Guilong Lu conceived and designed the experiments, performed the experiments, prepared figures and/or tables, authored or reviewed drafts of the article, and approved the final draft.

The following information was supplied regarding data availability:

No raw data was generated in this literature review.

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
