# Peer review of "Summer pruning in apple trees is an advisable cultural practice that promotes bud differentiation and improves fruit quality: a literature review"

_PeerJ, doi:10.7717/peerj.20229_

## Round 0.1 · original submission · Major Revisions

**Language Note:** The review process has identified that the English language must be improved. PeerJ can provide language editing services - please contact us at [email protected] for pricing (be sure to provide your manuscript number and title). Alternatively, you should make your own arrangements to improve the language quality and provide details in your response letter. – PeerJ Staff

Reviewer 1 ·

Basic reporting

The manuscript is generally well-written and structured. The language is accessible and appropriate for an international scientific audience, although minor grammatical polishing (e.g., removing inconsistent use of the singular/plural form, awkward phrases such as “obtains much higher fruit quality” in the Abstract, line 22) would further improve readability. The literature cited is extensive, relevant, and current, with sources spanning both international and Chinese-language databases, which strengthens the global relevance of the review. Figures are conceptually appropriate, and the survey methodology is adequately described.

Experimental design

The manuscript aligns with the aims and scope of a review article in an applied agricultural science journal. The review’s methodology appears rigorous and covers multiple databases and document types without geographic restriction. The authors appropriately exclude genetic studies to maintain focus on horticultural practice. However, the survey strategy would benefit from more systematic criteria for study inclusion/exclusion, and possibly a PRISMA-style diagram or summary table to improve reproducibility.

Validity of the findings

The conclusions are generally well-stated and flow logically from the reviewed content. The manuscript effectively synthesizes the physiological and agronomic impacts of summer pruning, supported by a broad base of literature (pp. 5–8). The authors also provide a reasonable discussion of the limitations and challenges in both manual and robotic pruning (pp. 8–9), though this could be expanded with more specific unresolved research questions and avenues for future work. The discussion of robotic pruning is novel and timely, but somewhat limited in depth; more technical or implementation-focused content (e.g., validation studies, performance metrics) would strengthen this section. Additionally, the review does not overstate findings, but occasionally makes general claims (e.g., “summer pruning improves the ventilation and light conditions of trees”) that should be qualified with context (e.g., pruning intensity, cultivar, or climate).

Reviewer 2 ·

Basic reporting

The English is acceptable. The introduction and background information are well justified by the reason for the review. Literature is generally well referenced. Structure conforms to PeerJ standards and also the discipline norm. There are several studies on summer pruning; however, the point of the review is well-focused, and it is quite useful. The introduction adequately introduces the subject and makes the audience and motivation clear. The figures are relevant to the content of the review. Summary tables of cited literature and main findings in each main section would help the Reader to see the current status of the topic.

Experimental design

Article content is within the scope of the journal. A rigorous and well-designed literature search was performed to a high technical & ethical standard. Methods are described with sufficient detail. The sources are adequately cited. The review is organized logically into coherent subsections.

Validity of the findings

The impact of summer pruning is assessed. The novelty of the study has to be emphasized more. Conclusions are shortly stated and linked to the original research question & limited to supporting results. The aims are fulfilled in the review text. The conclusion identifies future directions, but this section is very short and needs to be elaborated.

Additional comments

In case of plant protection L70-71 and L251-252: There are many more plant protection effects of pruning, including summer pruning, that also affect fruit quality. Pruning also has different effects in integrated and organic orchards, as growing ability is different in these orchards. Summer pruning also reduces the inoculum sources of certain pathogens (e.g., apple scab and powdery mildew).

---

## Round 0.2 · accepted · Accept

Dear Dr. Que, I congratulate you on the acceptance of this article for publication.

Reviewer 1 ·

Basic reporting

Yes, review of broad and cross-disciplinary interest

Experimental design

Survey Methodology consistent with a comprehensive, unbiased coverage of the subject

Validity of the findings

-

Reviewer 2 ·

Basic reporting

The study is generally improved. The English is acceptable. The introduction and background information well justify the reason for the review. Literature is generally well-referenced. Structure conforms to PeerJ standards and also the discipline norm. The introduction adequately introduces the subject and makes the audience and motivation clear.

Experimental design

Article content is within the scope of the journal. A rigorous and well-designed literature search was performed to a high technical & ethical standard. Methods are described with sufficient detail. The sources are adequately cited. The review is organized logically into coherent subsections.

Validity of the findings

The impact of summer pruning is well assessed. Novelty of the study is now well emphasised. Conclusions are shortly stated and linked to the original research question & limited to supporting results. The aims are fulfilled in the review text. The conclusion also identifies future directions.